# Which Utterance Types Are Most Suitable to Detect Hypernasality Automatically?

**Ignacio Moreno-Torres** [1,*] , **Andrés Lozano** [2], **Enrique Nava** [2] and **Rosa Bermúdez-de-Alvear** [3]

1. Departamento de Filología Española, Universidad de Málaga, 29071 Málaga, Spain
2. Departamento de Ingeniería de Comunicaciones, Universidad de Málaga, 29071 Málaga, Spain; ald@uma.es (A.L.); en@uma.es (E.N.)
3. Departamento de Personalidad, Evaluación y Tratamiento Psicológico, Universidad de Málaga, 29071 Málaga, Spain; bermudez@uma.es
* Correspondence: imoreno@uma.es

**Featured Application: The results of this study provide key information, both linguistic and technical, to develop a Spanish language hypernasality detection tool which could be used by non-experts and could run on universally available mobile devices. The results may also guide the development of hypernasality detection tools for languages other than Spanish.**

**Abstract:** Automatic tools to detect hypernasality have been traditionally designed to analyze sustained vowels exclusively. This is in sharp contrast with clinical recommendations, which consider it necessary to use a variety of utterance types (e.g., repeated syllables, sustained sounds, sentences, etc.) This study explores the feasibility of detecting hypernasality automatically based on speech samples other than sustained vowels. The participants were 39 patients and 39 healthy controls. Six types of utterances were used: counting 1-to-10 and repetition of syllable sequences, sustained consonants, sustained vowel, words and sentences. The recordings were obtained, with the help of a mobile app, from Spain, Chile and Ecuador. Multiple acoustic features were computed from each utterance (e.g., MFCC, formant frequency) After a selection process, the best 20 features served to train different classification algorithms. Accuracy was the highest with syllable sequences and also with some words and sentences. Accuracy increased slightly by training the classifiers with between two and three utterances. However, the best results were obtained by combining the results of multiple classifiers. We conclude that protocols for automatic evaluation of hypernasality should include a variety of utterance types. It seems feasible to detect hypernasality automatically with mobile devices.

**Keywords:** hypernasality; Spanish language; speech acoustic features; ANN; automatic detection of speech deficits

## 1. Introduction

Speech is described as hypernasal when there is an abnormal increase in nasal resonance during the production of oral sounds. This condition results from an insufficient closure of the velopharyngeal port that allows the air stream to flow through the nasal cavity during the production of oral vowels and consonants. It, thus, may lead to phonological error patterns in consonants (e.g., b > m, d > n) and/or to nasalized vowels (e.g., ã ẽ ĩ õ ũ). Hypernasality (HN) is commonly observed in patients with cleft palate (CP) and also in other groups of patients who have short velum which cannot achieve a complete contact with the posterior pharyngeal wall (e.g., those with a 22q11.2 deletion syndrome; 22q11.2DS). Therefore, evaluating HN is most relevant to make clinical decisions and to plan effective intervention in these patients [1].

Traditionally, HN has been evaluated perceptually (e.g., CAPS-A protocol; [2]). However, perceptual evaluation of HN is a complex task, particularly for those without a highly



specialized training in speech therapy (e.g., many otorhinolaryngologists, pediatricians or teachers). Part of the difficulty is due to nasality being a gradual phenomenon rather than a categorical one and to the fact that healthy speakers nasalize to some degree some oral sounds (as a result of poor coordination of the velum with other articulators, or to trans-palatal transmission of acoustic energy from the oral cavity to the nasal one [3]). In addition, in many languages such as Spanish, nasal vowels do not exist as phonemes, which means that most speakers have difficulties in recognizing nasal vowels as distinct categories. Finally, pathological HN often co-occurs with compensatory errors and/or low intelligibility, which may make it even harder to identify it [4]. These considerations have motivated researchers to develop objective measures of HN. One promising approach consists in using automatic classification systems trained with different sets of acoustic features [5]. This approach has two major advantages: it is non-invasive and the required technology is nowadays universally available (e.g., by using mobile phones).

In the past there have been many proposals to evaluate HN based on acoustic information. While the results in terms of accuracy are generally excellent, most studies have used only a limited number of utterance types, such as sustained vowels [6–11]. This is not clearly compatible with standard clinical recommendations [2], which strongly recommend that patients are evaluated using a variety of phonemes and utterances with varying complexity (e.g., CAPS-A protocol, [2,12]).

As regards phonemic diversity, the CAPS-A protocol identifies three levels of nasalization, depending on which segments are nasalized: (1) mild: nasalization evident only on closed vowels (e.g., /i u/); (2) moderate: nasalization observable in closed and open vowels (e.g., /a e o i u/); and (3) severe: nasalization observable in all vowels and in voiced consonants (e.g., /b d g/). Furthermore, Kummer et al. [12] proposes that syllable series including the voiceless stops such as /p t k/ should be used to study nasalization. As regards to utterance complexity, it is generally emphasized that different utterance types are needed to provide valuable information (e.g., isolated vowels, words, sequences of syllables repetition, sentences and spontaneous speech [2,12–16]). However, Kummer et al. [12] consider that two of these tasks are especially helpful: on the one hand, repetition of syllabic sequences such as /ta ta ta ta ta . . . /, which allow to isolate individual phonemes and eliminate context effects; on the other, sentences that contain multiple productions of the same phoneme placement, which allow to assess the presence of nasal emission in a connected speech environment. To summarize, according to clinical experts, evaluation of HN should be based not only on isolated vowels but also on a variety of voiced and unvoiced consonants which are combined in different utterance types.

Acoustic based studies have used two different approaches to detect HN. Mathad et al. [4] used Automatic Speech Recognition technology (ASR) to determine which sections of an utterance have been nasalized. To this end the authors trained an ASR system to classify audio segments as nasal-vowel, nasal-consonant, oral-vowel, or oral-consonant. While the results of this approach are most promising, it is important to note that it requires access to large, annotated corpora that are available only for a few languages and only for adult speech, which limits its applicability.

A second approach consists in using classification algorithms trained with acoustic features of specific fragments of a speech signal. Classification algorithms include Random Forest (RF), Support Vector Machine (SVM) or Artificial Neural Network (ANN). Acoustic features used in previous studies include, among others, Mel Frequency Cepstrum Coefficients (MFCCs), the Voice Low Tone to High Tone Ratio (VLTHTR) and the vowel formants and their bandwidth [6–11]. In most studies using this approach the fragments analyzed were either sustained vowels [6,8] or vowel fragments that had been annotated manually in words or sentences [7,9–11]. Only a few studies have used complex utterances to automatically evaluate nasality [17–19]. Golabbakhsh et al. [18] used six sentences containing stop and fricative consonants which are used routinely by speech therapists to perceptually assess the quality of speech. The authors trained an SVM classifier with a pool of acoustic features (e.g., jitter, shimmer, MFCC, bionic wavelet transform entropy and bionic wavelet

transform energy) which were computed for each utterance. In the best case, accuracy reached 85% with a sensitivity of 82% and a specificity of 85%. Orozco-Arroyave et al. [17] analyzed a database of 108 healthy and 128 hypernasal Spanish speaking children. All the children produced the five Spanish sustained vowels and two words, one with unvoiced consonants (i.e., *koko*) and one with one voiced and one unvoiced consonant (i.e., *gato*). They trained an SVM classifier using non-linear dynamics features along with a set of six entropy measures. The results were the best for the vowels /a, i, e, o/ and for the word *gato* (i.e., the one with a voiced consonant); the poorest results were observed with vowel /u/ and word *koko*. However, the results improved when they selected the best features from each vowel (accuracy 91%; sensibility: 93–95%; specificity: 88–90%). Altogether these results indicate that it is feasible to evaluate nasality automatically using running speech and also by combining different speech samples from the same speaker.

To summarize, there seems to be a mismatch between clinical studies, on the one hand, which emphasize the importance of exploring a variety of speech sounds and utterance types and automatic analysis research, on the other, which has focused mainly on sustained vowels or a limited number of words or sentences. One obvious reason why most technical studies have used sustained vowels is because in such case the spectrum is stable throughout a relatively long window, which increases the probability of detecting the relatively small acoustic effects introduced by the nasal resonance. If more complex utterances are considered (e.g., full sentences) and assuming that we do not use an ASR approach, it will be necessary to compute the average spectrum, which may blur the local effects of nasality. However, as shown by Orozco-Arroyave et al. [17] and others, at least in some cases the average spectrum may serve to detect HN. Indeed, it seems reasonable to speculate that the effects of HN might be measurable in the same utterance types in which humans perceive HN with relative ease (e.g., repeated syllables, sentences with voiced consonants, etc. [12]) and that the results might be improved by combining multiple utterances. It remains to clarify to what extent these speculations can be confirmed.

This study explores to what extent utterances other than sustained vowels can be used to detect HN automatically. Our main aim is to identify which utterances or combination of utterances, if any, might be the most optimal ones for this task. We expected that, similarly to what has been observed in clinical practice, the effects of HN might be most clearly observable in some subtypes of utterances (e.g., repeated syllables, words and sentences [12]). In the long term, we aim to develop a HN detection tool that is accessible to a large audience and particularly to clinicians without speech therapy expertise (e.g., pediatricians, otorhinolaryngologists) In order to cope with this long-term objective, we decided to collect the data using a mobile app. Thus, a second aim of this study is to test to what point audio data obtained using a mobile app can be used to evaluate HN. We anticipated that, thanks to the advances in mobile phone devices, it might be possible to detect HN.

## 2. Materials and Methods

As part of this study, we created a database of healthy and hypernasal Spanish speakers. The speakers were either children or female adults. The data were obtained using a mobile app, ASICA (see Appendix A for instructions), developed as part of this project, which allowed us to include patients from three different Spanish speaking countries (Chile, Ecuador and Spain). Note that this mobile app was developed as a response to the COVID-19 crisis and the impossibility of recording the participants in our lab. Based on these recordings we proceeded to define a hypernasal dataset and an oral dataset. For this end, two approaches might be considered. One consists in including in the hypernasal dataset only those utterances for which there was perceptual evidence of HN (i.e., item-by-item approach). Another consists in including in the hypernasal dataset all the utterances of those speakers which have been previously classified as hypernasal (i.e., speaker-by-speaker approach). Given the difficulty to annotate the full database item-by-item, we decided to use a speaker-by-speaker approach.

The resulting datasets were used to run multiple tests that consisted in training and evaluating a HN classifier. Each test was characterized by: (1) the utterance or list of utterances used to train and evaluate the classifier (e.g., syllable sequence *ta ta ta*, the word *dedo*, both the sequence *ta ta ta* and the word *dedo*); (2) the subset of acoustic features (e.g., MFCC4 of utterance *tatata* ... ) and the classification algorithm (e.g., RF, SVM or ANN). As a first step we ran forty-four tests, each one with one utterance in the database. We assumed that as the accuracy of the automatic classifiers would be partly determined by the utterances used to train them, when accuracy was very high (e.g., "in discriminating healthy speakers from patients by using syllable repetition"), it would indicate that the utterance used was an optimal candidate to automatically evaluate HN. Next, we ran tests using more than one utterance to train and evaluate each classifier. For this end we created what we call optimal lists of utterances (see details in Section 2.6). Finally, we explored the feasibility of computing, for each speaker, a hypernasality score (HN Score) by combining the results of multiple tests of the same participant. Note that this last approach emulates the scoring used in many speech evaluation tasks, in which a partial score is provided per item and a global score is obtained by computing (e.g., summing) the results of the individual items in the task.

### 2.1. Database and Selected Participants

The database was created with the help of an IOS app named ASICA. It included the recordings of 54 patients and 49 healthy speakers, all of whom were native Spanish language speakers. In this database we defined as patient any speaker with a clinical history associated to the presence of HN, which means that some patients did produce hypernasal speech when they were recorded, whereas others were non-nasal due to previous successful treatments. The patients were from Spain (N = 36), Chile (N = 16) and Ecuador (N = 2) and they were recruited from diverse clinical facilities and parents' associations:

- Hospital Materno infantil de Málaga, Spain (N = 12);
- ASAFiLAP, the Andalusian Cleft Palate Association, Spain (N = 9);
- 22q.11 Andalusian Association, Spain (N = 10);
- Fuensocial CAIT Fuengirola, Spain (N = 1);
- Clínica Médica Fuengirola, Spain (N = 3);
- Independent Speech Therapist, Spain (N = 1);
- Hospital Gantz, Santiago de Chile (N = 16);
- Independent Speech Therapist, Ecuador (N = 2).

The control speakers were recruited through social media and with the help of the patients' associations. For the purpose of this study, we selected from the database those patients that matched these criteria:

1. Age and sex: female adult 18–42 years old or child 5–15 years old.
2. Mean fundamental frequency: above 180 Hz.
3. Data completion: the patient produced, at least, 90% of the utterances in the task.
4. Audio quality: loud masking noise was observed in fewer than a 10% of the utterances.
5. Confirmed HN. In the case of the patients, HN was confirmed perceptually by three trained speech therapists in at least five utterances.

The application of these criteria resulted in a total of 39 patients (1 from Ecuador, 15 from Chile and 23 from Spain). The control group was selected so that it included the same number of speakers as the experimental group, with the two groups matched on age (for children) and age and sex (for adults). All the control speakers were from Spain except one that was from Ecuador.

### 2.2. Materials

The protocol to collect speech samples was elaborated according to the International Speech Parameters Group recommendations [14]. It is a double aimed protocol since it pretends to obtain a list of utterances that is sufficiently informative so as to perceptually

evaluate HN and, furthermore, it provides reliable outcomes that can be compared with other studies, independently of the language of testing [2,12,13,15,16]. It contains six subtasks (T1–T6) and registers three types of speech samples: (1) rotten speech is recorded by counting from 1 to 10 (T1); (2) repeated speech is obtained by means of sequences of syllables (T2); words (T5); and sentences (T6); and (3) sustained sounds are represented by two fricative consonants /f, s/ (T3) and vowel /a/ (T4). Table 1 shows the list of utterances in each subtask as well as the instructions provided to the participants.

**Table 1.** Utterances in the repetition task.

| Subtask | Instruction | Utterances |
|---|---|---|
| T1. Counting | Count one to ten | *Uno, dos, tres, cuatro, cinco seis, siete, ocho, nueve, diez* |
| T2. Syllables | Repeat the syllable rapidly | *pa pa pa . . . , ta, ta, ta . . . , ka ka ka . . . pi pi pi . . . , ti, ti, ti . . . , ki ki ki . . .* |
| T3. Sustained consonants | Produce a long consonant | */fffff . . . / /sssss . . . /* |
| T4. Sustained vowels | Produce a long /a/ | */aaaaa . . . /* |
| T5. Words | Imitate these words | *moto, boca, piano, pie, niño, llave, luna, campana indio, dedo, gafas, silla, cuchara, sol, jaula, zapatos* |
| T6. Sentences | Imitate these sentences | Voiced stop consonants: *Al gato de Ágata le gusta el yogur (/g/) A David le duele el dedo (/d/). El bebé va bien con babuchas (/b/)* Voiceless stop consonants: *Tómate toda tu taza de té (/t/) Papá puede pelar a Pili (/p/) Quique coge el papel de calco (/k/)* Fricative consonants: *Si llueve le llevo la llave a Yolanda (//) Susi sale sola y se ensucia (/s/) Fali fue a la feria inflando un globo (/f/) Los zapatos de Cecilia son azules (/θ/) La jirafa de Jesús se mojó jugando (/x/)* Affricate consonant: *Chuchu y Chelo chillan mucho (//)* Approximants: *Lali y Luna leen los carteles (/l/)* Vowels: *Uy, ahí hay algo* Nasals: *Mi mamá me mima mucho (/m/) El nene nos canta una nana (/n/)* |

The syllable repetition task (T2) consists of a series of consonant–vowel sequences. Three types of plosive consonants (i.e., /p t k/) are used to construct this task because their articulation pattern requires good velar motor coordination between the unvoiced occlusion and the vowel. In order to avoid a too long repetition task, which is quite boring for kids, only two vowels are used to design these syllables sequences. The vowel /i/ was selected because it is the one that requires the softest velar closure (softer than other closed vowels such as /u e o/); and the vowel /a/ was selected because it requires a quite harder velopharyngeal closure. As regards to the words repetition task (i.e., T5), 16 items have been selected from a previously published test [20]. Selection criteria consisted of gathering a representation of most of the Spanish consonants in a Consonant-Vowel context. Syllables with complex onset (e.g., *pl*, *pr*, as in *pla*, *pra*) were avoided because of their motor complexity, which requires greater involvement of articulators such as tongue or lips

and, thus, they deserve a linguistic maturity that falls beyond the velopharyngeal closure capability. The T6 subtask includes 16 sentences, each one with predominance of one type of consonant. Following international recommendations, two sentences with nasal consonants were also included [14].

### 2.3. Data Collection

In order to participate in the study, the participants used the mobile app ASICA, which can be downloaded from Apple Store. Before running the app each participant (or parent/tutor in the case of minors) was instructed regarding the aim of the study and was required to sign one informed consent form. Then, the participants were advised to watch a short 2 min video that illustrates how the task proceeds and gives some advice regarding the need to avoid background noise; when necessary, further details were given by phone or email. Finally, each participant was provided a unique ID that is necessary to run the app.

Once the user opens the app, he or she is requested to insert the ID. The task starts with one example token after which the six subtasks described in Table 1 are run sequentially. Each task begins with a short description indicating the type of utterance that is going to be presented and continues with the utterances to be imitated. For each utterance, the app produces the utterance and then waits between 3 and 9 s (depending on the target utterance) for the speaker to imitate it. Once the task is completed all the audio recordings are stored in a cloud database from where our research team downloaded them for further analyses.

### 2.4. Acoustic Features

Due to the remote and unsupervised nature of data collection and the wide range of patients' characteristics, the recording was designed to last longer than usual. Hence, audio data will contain a substantial amount of silence before and after the task-objective audio, introducing background noise of no interest in the classification analysis. In order to reduce the amount of silence recorded on each task we used, before feature calculation, a speech/background discriminator based on [21]. The discriminator, which assumes that the first 100 ms contain background noise, computes the endpoints of each speech utterance.

Based on previous evidence, the following selection of features were computed from each utterance: MFCC coefficients, the first three formants together with their bandwidths (BW) and distances and the VLTHTR [6,10,22,23]. The 13-dimensional MFCC features were calculated using moving Hamming windows. The windows were 25 ms long with 15 ms overlap. The first MFCC coefficient (MFCC0) was computed as the log energy of the signal. Delta and delta-delta MFCC values (first and second derivatives) were not included because they showed a low capacity to differentiate healthy and hypernasal speech during initial analyses. A set of F1, F2 and F3 formants was computed using 16-order linear prediction coefficients (LPC) together with their bandwidth, which was also used to calculate the distances among the formants (i.e., F1-F2, F1-F3 and F2-F3). For all these features, a mean value was obtained for each utterance. VLTHTR was defined as the logarithmic ratio between the signal power at low and high frequencies [6]. The audio spectra were derived using the long-term average spectrum calculated from the average power spectral density obtained from a series of overlapping FFTs; the FFT length was 4096 and the hop size was 2048. In this study, multiple cutoff values were used so separate low and high frequencies: from 400 Hz to 900 Hz with 100 Hz steps. All these features were calculated using custom code in Matlab R2020b and Audio Toolbox functions.

### 2.5. Classification Algorithms

We evaluated the performance of three different methods for classifying the audio data: RF, SVM and ANN. In all these methods, a feature selection and reduction process was conducted before classification, using an absolute value two-sample *t*-test with pooled

variance estimate algorithm. Each method was evaluated independently using (K = 5)-fold cross-validation. This method ensures that each fold of the classifier has equal proportion of data from each class and allows to reduce any error due to partition of the data. Finally, note that the feature selection and reduction process was repeated for each fold, which ensures a neat separation between the training and testing processes.

The SVM classifiers used linear kernel function with auto-optimization of hyperparameters. The RF classifiers used bootstrap-aggregated decision trees. The architecture of ANN HN model is shown in Figure 1. The input layer consists of 20 nodes, corresponding to the 20-dimesional speech features selected for each classification task. The model is comprised of 4-hidden layers, where each layer has 1024 hidden neurons with rectified linear unit (ReLU) activation. The final output layer has 2 SoftMax nodes, each corresponding to one class of speakers (i.e., hypernasal and healthy).

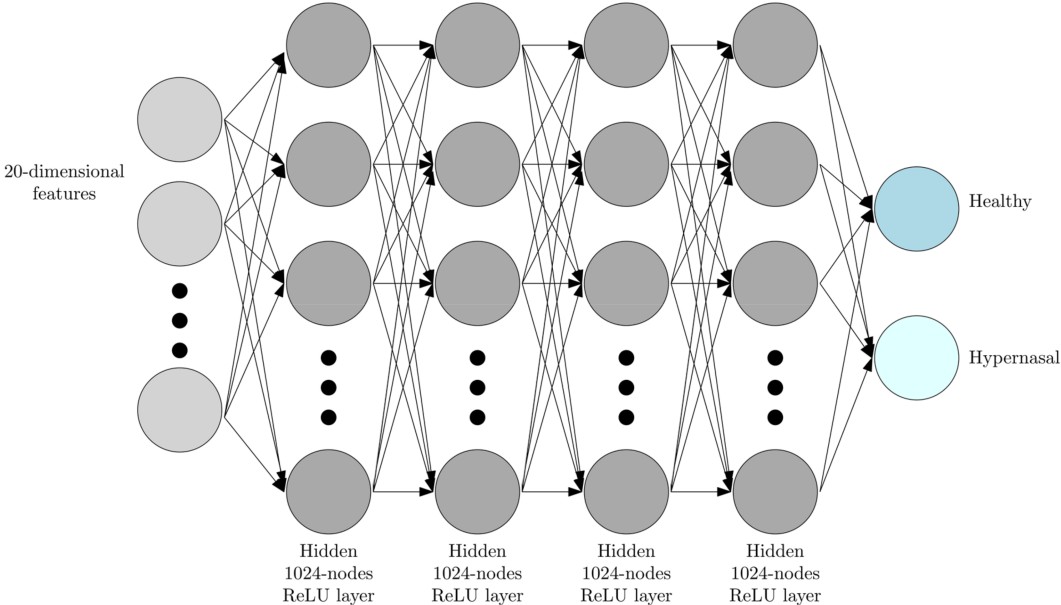

**Figure 1.** Architecture of the ANN model.

### 2.6. Utterance Selection and HN Score

As a first step we carried out forty-four single utterance tests (i.e., by using one utterance in each test). Next, we ran diverse tests in which each classifier was trained with multiple utterances (rather than a single utterance). As the number of possible utterance combinations is very high, the following procedure was used to create what might be an *optimal list of utterances*: (1) A (size = 1) list consisting of the utterance providing the highest accuracy in the one-utterance tests (i.e., *base list*) is created (e.g., word *dedo*); (2) a new test is run for each utterance which is not currently included in the base list (i.e., forty-three in the first step); in these tests, training and evaluation use the *base list* together with one new utterance (e.g., word *dedo* + syllable sequence *ka*; word *dedo* + syllable sequence *ki*, etc.); the utterance in the test with the highest accuracy is added to the *base list*; (3) step 2 is repeated while the accuracy increases. This process is repeated separately for SVM, RF and ANN.

Finally, we explored the feasibility of computing a hypernasality score (HN Score) for each speaker. The HN Score was the result of dividing the number of times that the speaker has been classified as HN by the total number of tests. Thus, the HN Score ranges from 0% (never classified as HN) to 100% (always classified as HN). Note that the HN Scores may vary depending on the actual tests used to compute it, for which it will be necessary to determine which list of tests provides the best results (i.e., to discriminate the patients from the healthy speakers). In order to interpret the results obtained with the HN Score it is important to consider that some participants may score close to 50% (i.e., indicating

that the HN Score does not clarify whether or not the speaker is HN). Here, we use these criteria to interpret the results:

- HN Score < 40%: the speaker is not HN;
- HN Score in the range 40–60%: HN can be neither confirmed nor discarded;
- HN Score > 60%: the speaker is HN.

## 3. Results

### 3.1. Preliminary Analyses of the Database

One of the aims of this study was to explore the feasibility of using mobile devices to assess HN. Thus, it is important to examine the causes to exclude 15 out of 54 patients in the database. The causes for exclusion were the following:

- Four patients produced fewer than 90% of the utterances. Two of them were three years old and two more were four years old.
- One participant was excluded due to background noise masking his utterances.
- One male participant had a mean F0 of 156 Hz.
- HN was not confirmed in nine patients. These patients were non-nasal due to previous successful treatments.

It is important to highlight that the selected database was far from ideal: many selected audio recordings had background noise and some speakers did not complete the full task. As regards to ambient noise, we manually annotated the utterances for which there was background noise above 50 dB (with the audios normalized to 70 dB). For most speakers 50 dB background noise was not frequent (i.e., occurring in fewer than 10% of the utterances). Noise was frequent (i.e., >20% of the utterances) in 14% of the controls and 36% of the patients. As regards to data completion, 86% of the selected controls and 63% of the selected patients produced all the utterances and only one control and one patient failed to repeat more than five utterances. Figure 2 shows two illustrative examples of recording with and without background noise.

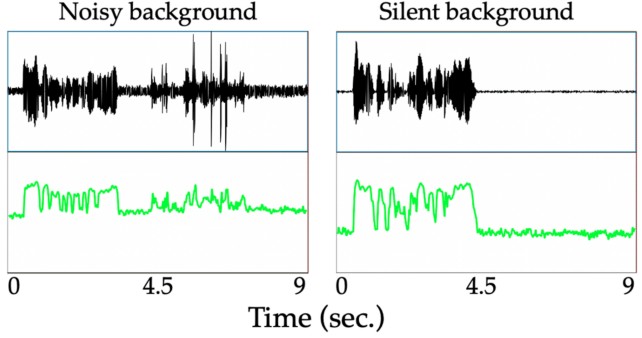

**Figure 2.** Audio sample with background noise (speaker fis06825) and with silent background (speaker fis09874). Both samples are normalized to 70 dB. The black line represents audio data. The green line is the absolute value of amplitude of the signal.

Finally, in order to have a better understanding of the patient's data, we decided to annotate item-by-item the utterances for which there was at least one instance of nasalized consonant. Note that this is not a full description of the database, because vowels were not manually annotated. However, this could provide a clear idea regarding the extent of the variability within the database. As expected, the results were highly variable: no child nasalized (one or more) consonants in all the utterances and no utterance was nasalized by all the patients (see Figure 3). For instance, the patient that nasalized the most (i.e., 813) did not nasalize the consonants in several utterances that were nasalized most frequently (e.g., *pa pa pa . . . , boca, silla*, etc.)

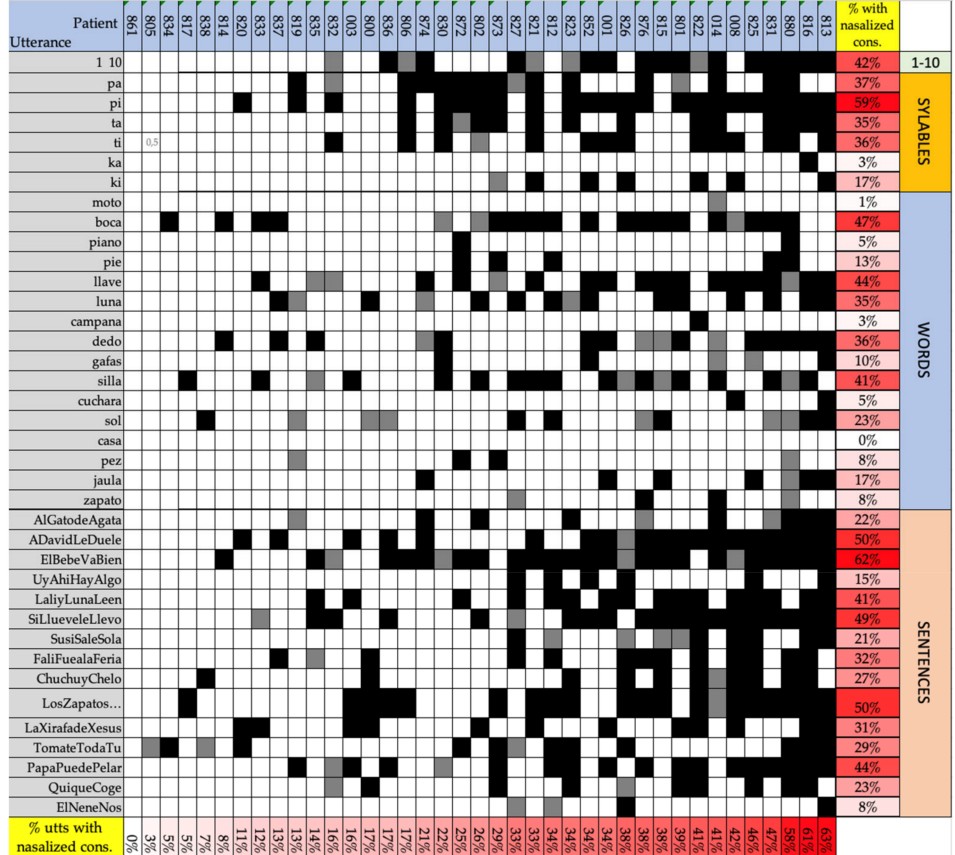

**Figure 3.** Item-by-item detail of the utterances with at least one consonant nasalized. Each column corresponds to one patient and each row to one utterance in the repetition task. Black cells: two experts agree that at least one oral consonant has been nasalized in the utterance. Grey cells: the experts disagree. White cells: two experts agree that no oral consonant has been nasalized. Note that this table is an incomplete description of the database: it does not show vowel nasalization.

### 3.2. Results for Forty-Four Single Utterances

Figure 4 shows the accuracies obtained for the single-utterance tests. The values ranged between a minimum of 46% (ANN-T5 luna) and a maximum of 81% (SVM-T5 dedo). The best results were obtained with SVM classifiers in 25 utterances, with RF in 17 utterances and with ANN in only 8 cases. As Figure 4 shows, the best results (i.e., accuracy >75%) were obtained with four of the syllable sequences, two words and one sentence. Then, there is a group of utterances for which the accuracy was also relatively high (i.e., 70–75%) and which included the two other syllable sequences, one word, five sentences and the sustained consonant /f/. The lowest scores (i.e., 50–60%) were obtained with eight of the words and one sentence. Altogether, these results show that the accuracy of automatic classifiers varies substantially depending on the utterance used to train them.

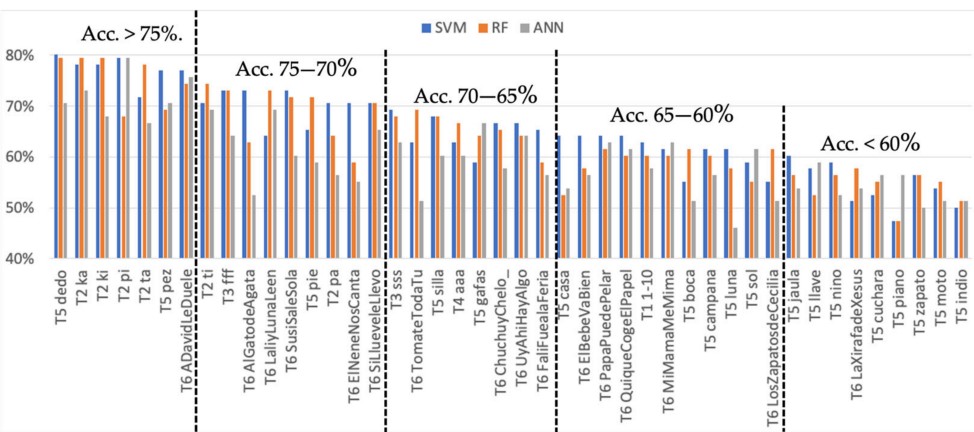

**Figure 4.** Accuracy of the 44 single-utterance SVM, RF and ANN tests. The utterances are sorted, from left to right, according to the accuracy in the best classifier.

### 3.3. Multiple Utterance Training Corpus

Following the procedure described in the Section 2.6, we computed the optimal list of utterances for each algorithm (see Table 2). In the case of the RF and ANN tests, the results showed an increase in accuracy from one to two utterances and a reduction with three or more utterances. In the case of SVM, the peak accuracy was obtained with three and four utterances. As it is possible that our approach might miss better candidate utterance lists, we decided to run further tests using the N most accurate utterances (for N from 2 to 12). For instance, in the case of the SVM classifier, we ran tests with these utterance lists: (1) *T5 dedo*; (2) *T5 dedo + T2 pi*; (3) *T5 dedo + T2 pi + T2 ka*, etc. None of these tests resulted in accuracy rates higher than the ones obtained with the optimal lists presented in Table 2.

**Table 2.** Optimal utterance lists.

| Algorithm | Optimal Lists and Cumulative Accuracy |
|:---:|:---:|
| SVM | Base list: T5 dedo (81%)<br>+T3 fff (88%)<br>+T2 pa (92%)<br>+T6 Susi sale sola (92%)<br>T6 A David (91%) |
| RF | Base list: T2 ka (84%)<br>T5 dedo (86%)<br>(2) +T3 f (83%) |
| ANN | Base list: T2 pi (79%)<br>+T6 A David (86%)<br>+T2 ka (76%) |

### 3.4. Combining Classifiers: HN Score

As explained in the Method section, by combining the results of multiple utterances we obtained a HN Score per participant. In order to facilitate the interpretation of these results we assume that HN Score > 60% indicates that the HN has been confirmed, while HN Score < 40% indicates that it is rejected; finally, scores in the 40–60% range do not allow to confirm or discard HN. Figure 5 shows the results obtained when using: (1) forty-four classifiers (i.e., one per utterance; HN Score (44)); note that for each utterance we chose the classifier with the highest accuracy (i.e., SVM, RF or AMM),(2) the sixteen classifiers with accuracy above 70% (i.e., HN Score (16)) and (3) the seven classifiers with accuracy above 75% together with the optimal SVM list shown in Table 2 (i.e., HN Score (7 + *Sel*)). In all three cases, the majority of the patients scored higher than 60% and the majority of the controls scored lower than 40%. However, there were some clear differences between the three selections.

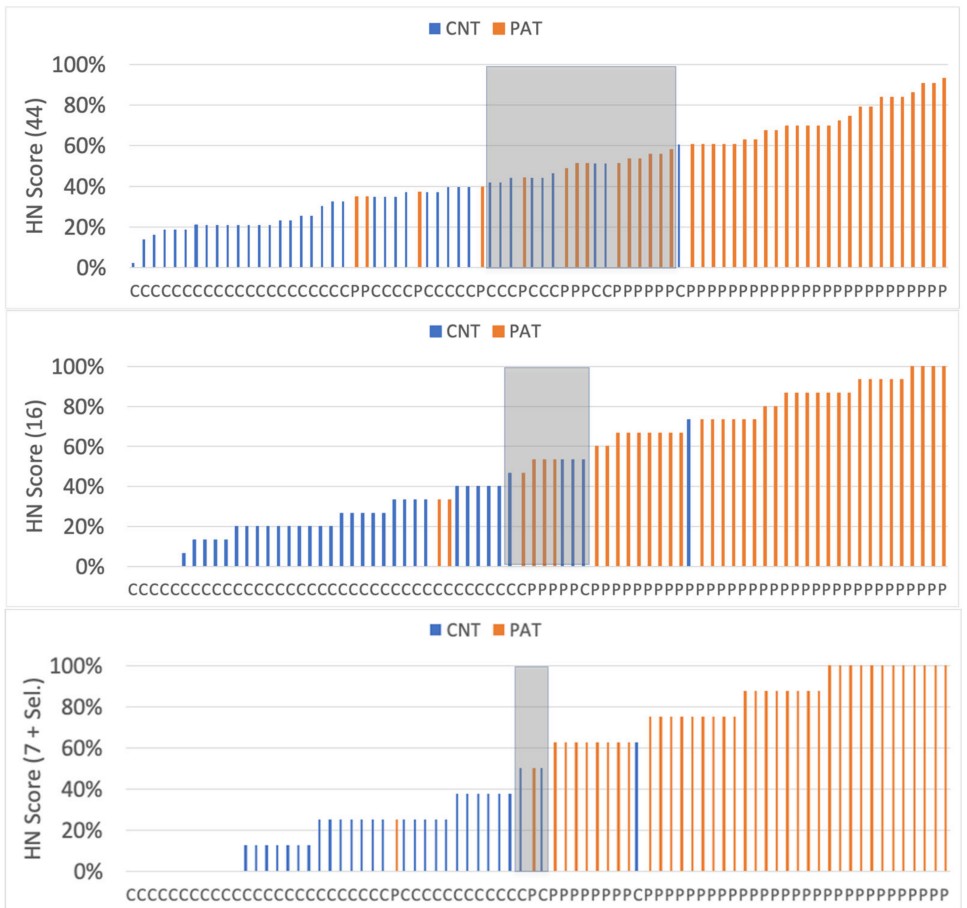

**Figure 5.** HN Score using forty-four single-utterance (**top**), using the best 16 utterances (i.e., with accuracy > 70%) (**middle**), and using the best 7 utterances + the SVM optimal list (**bottom**). Each blue bar represents one control participant. Each orange bar represents one patient. The grey box shows the speakers with intermediate scores (i.e., 40–60%). Note that the blue bars to the left of the grey box and the orange bars to the right are, respectively, true negatives (TN) and true positives (TP). On the contrary, the orange bars to the left are false negatives (FN) and the blue bars to the right are false positives (FP).

The group mean for the patients was 66% when using the forty-four one-utterance classifiers (HN Score (*44*)) and it increased to 77% in HN Score (*16*) and to 81% in HN Score (*7 + Sel*). In the case of the controls the mean values were, respectively, 31%, 24% and 20% (for HN Scores *44, 16* and *7 +Sel*). The increased discriminability of HN Score (*7 + Sel*) can also be observed by examining the results qualitatively (Figure 5 bottom): In this case there were only three speakers in the 40–60% region (Figure 5 bottom) and only one false negative (i.e., a patient scoring below 40%) and one false positive (i.e., a control speaker scoring above 60%). Thus, the discriminability improves as the number of tests used to compute the HN Score was reduced. Table 3 provides some details of the utterances and features used in the test selected for HN Score (*7 + Sel*). Note that in Table 3 the list of features included in each case are those which were selected in the five folds during the cross-correlation train-evaluation process (see Section 2).

**Table 3.** Utterances and classifiers used in the HN Score (*7 + Sel*).

| Utterances | Alg. | Results | Features |
|---|---|---|---|
| *T5 dedo*<br>*+T3 fff*<br>*+T2 pa* | SVM | Spec.: 92%<br>Sens.: 92%<br>Acc.: 92% | *T5 dedo*: bwf1, f3, f1-f3, f2-f3<br>*T3 fff*: mfcc2, mfcc10, fcc13<br>*T2 pa*: mfcc1, mfcc5, mfcc9 |
| *T5 dedo* | SVM | Spec.: 85%<br>Sens.: 77%<br>Acc.: 81% | bwf1, bwf2, f3, f1-f3, f2-f3<br>mfcc2, mfcc5, mfcc11, mfcc12<br>vlhr400, vlhr500, vlhr600, vlhr900 |
| *T2 ka* | SVM | Spec.: 73%<br>Sens.: 85<br>Acc.: 79% | bwf1, bwf3, f1, f3, f1-f3, f2-f3,<br>mfcc1, mfcc3, mfcc4, mfcc6, mfcc7, mfcc10,<br>mfcc11, vlhr900 |
| *T2 ki* | RF | Spec.: 75%<br>Sens.: 83%<br>Acc.: 79% | bwf1, bwf2, bwf3, f1,<br>mfcc1, mfcc2, mfcc3, mfcc4, mfcc6, mfcc7,<br>mfcc11, mfcc12, vlhr800, vlhr900 |
| *T2 pi* | RF | Spec.: 79%<br>Sens.: 79%<br>Acc.: 79% | bwf1, bwf3, f2, f3, f1-f3, f2-f3<br>mfcc1, mfcc2, mfcc3, mfcc5, mfcc6, mfcc8,<br>mfcc12, vlhr600, vlhr700 |
| *T2 ta* | RF | Spec.: 75%<br>Sens.: 80%<br>Acc.: 77% | bwf1, bwf2, f3, f1-f3, f2-f3<br>mfcc1, mfcc2, mfcc4, mfcc5, mfcc6, mfcc7,<br>mfcc8, mfcc9, vlhr400, vlhr500, vlhr600 |
| *T5 pez* | SVM | Spec.: 77%<br>Sens.: 77%<br>Acc.: 77% | bwf1, bwf3, f3, f1-f3, f2-f3<br>mfcc2, mfcc4, mfcc5, mfcc6, mfcc8<br>vlhr400 |
| *T6 A David* | SVM | Spec.: 77%<br>Sens.: 74%<br>Acc.: 76% | bwf1, bwf3, f2, f1-f2, f2-f3,<br>mfcc2, mfcc4, mfcc5, mfcc7, mfcc11,<br>mfcc12<br>vlhr400, vlhr500, vlhr600 |

Given that the controls' and the patients' recordings differed in the amount of background noise, we analyzed whether this had any effect on the HN Score. For this end, we divided the patients into two subgroups: those with background noise (N = 14) and those without background noise. Note that if the presence of background noise biased the results the patients with background noise might be classified as hypernasal more frequently that the remaining patients. However, the mean HN Score (*7 + Sel*) was slightly lower for the patients recorded with background noise than for those without noise (79% vs. 83%). Thus, it seems that the presence of noise did not bias the results.

## 4. Discussion

Two were the main objectives of this study. In the first place we aimed to determine whether or not it was possible to detect HN with utterances more complex than sustained sounds and, in that case, which were such utterances. A second aim was to test the feasibility of using mobile devices to detect HN. To this end we created a database of speech samples obtained with the help of a mobile app. From this database we excluded the patients for which there was no evidence of HN. It is important that the requisite to include the patients (i.e., individual-by-individual basis) implies that some of the utterances in the nasal group might be non-nasal. Given that this methodological decision might be considered a potential limitation we will begin the discussion with this issue. Then, we will discuss the main results.

The decision to select the participants on an individual-by-individual basis, rather than utterance-by-utterance, was motivated by practical considerations: it is clearly less time-consuming than the alternative approach. Our approach might have reduced the accuracy of the classifiers because many non-nasalized utterances were included in the patient's database. However, the relatively good results in terms of accuracy indicate that the approach was a valid one. Two factors may have contributed to our results. One is the

possibility that patients' HN persists in some utterances to such a minimal degree that the human ear is not capable of recognizing it. The other is that, as noted in clinical research, there are contexts which favor HN and the feature selection and reduction process may have discarded the utterances that do not favor HN.

Our results show that syllables sequences and, to a lesser extent, specific sentences, words and sustained consonants, are the most appropriate types of utterances to evaluate HN automatically. In contrast, the accuracy obtained by using rotten speech (i.e., counting one to ten) or sustained vowels is relatively low. In order to explain these results it may be helpful to compare the accuracy of three groups of utterances: (1) the /pa ta ka/ series versus the sustained vowel /a/; (2) rotten speech versus sentences; and (3) the sustained vowel /a/ versus the sustained consonant /f/.

As to first pair (i.e., syllable sequences vs. sustained vowel /a/), our results showed that the accuracy was relatively high in syllable sequences /pa ta ka/ and relatively low in /a/. This result indicates that the sustained vowels were very similar in the two groups, but the syllables were relatively different. In order to interpret these results, it is relevant to note that in the /pa ta ka/ series the same vowel is produced repeatedly; this means that, when produced by a healthy speaker, the average spectrum should be very similar to that of a vowel /a/. Thus, we interpret that patients are producing atypical syllable sequences. One possible interpretation of this result is that, due the increased effort required to produce the syllable sequences, the patients may struggle to control the velum, which may result in some degree of HN ([12]). However, it is also possible that the velopharyngeal insufficiency has led patients to modify slightly the articulatory patterns to produce these sounds (i.e., subtle compensatory mechanisms): these articulatory changes might modify the spectral configuration of the target sound in ways that may pass undetected to the human expert but that could be detected by the automatic classification system [24].

As to the second pair, rotten speech (i.e., counting one-to-ten) versus sentences, the accuracy of the former is clearly lower than that of some sentence (see Figure 4). Two factors may have contributed to this result. One is that numbers may have been practiced intensively by some children, for which at least some participants might be particularly effective in avoiding HN in this precise case. In contrast, the sentences may favor HN because they have multiple instances of the same consonant in different verbal contexts, all of which can impede the effective control of the velar closure. Another factor is that in the one-to-ten series, HN may occur occasionally (e.g., in one or two phonemes) and, thus, it may be blurred after averaging multiple window frames. In contrast, the phonological structure of some sentences may lead to relatively frequent instances of HN.

As to the sustained consonant /f/, the results indicate that this utterance might be effective in discriminating the patients from the controls either alone or in combination with other utterances (see Table 3). This result is relevant because, in the clinical context, this consonant is used commonly to detect air scape in HN patients, but not to detect nasal resonance. However, the results shown in Table 3 indicate that the two groups differed significantly in at least three MFCC features, suggesting that there are spectral differences between the /f/ sounds produced by the patients and the ones produced by the controls. Two possible explanations can be suggested for this result, which are identical to the ones noted in the case of the syllable sequences: the effect might be caused by presence of nasal resonance or, alternatively, it might be associated with learned articulatory patterns aimed to compensate the difficulty to generate sufficient oral air pressure.

Another relevant outcome of this study is that classifiers trained with between two and three utterances were more accurate than those using just one (see Table 2). This result extends those of Orozco-Arroyave et al. [17], who showed that classifying HN using individual vowels was less effective than using multiple vowel utterances. One possible explanation for these results is that HN is a complex phenomenon and that a single utterance type may not be sufficient to capture all the variation that can be observed among hypernasal patients. Note that the reduction in accuracy when the number of utterances increases is most possibly the result of the feature selection process, which did not take

into account the correlation between the selected features. This may have led to select redundant features and to discard other features that might contribute to the classification problem. However, our results also show that a better approach to capture the complexity of HN consists in combining the results of multiple classifiers. Using the HN Score we observed that combining the results of a small list of highly accurate classifiers allows not only to discriminate the two classes, but also to discard speakers for which the automatic classifiers provided conflicting results. This result shows that it is more effective to evaluate HN based on multiple utterances than based on a single utterance.

A secondary aim of this study was to determine the feasibility of using mobile devices to detect HN automatically. This approach presents at least two potential limitations: one related to the participants' implication in the task and another related to the acoustic context. Regarding the participants' implication, this may arise due to multiple circumstances (limited attention, lack of motivation, etc.) However, the results are promising as they show that the majority of the participants completed the task. The only exception were children aged three or four, for whom the task may have been too difficult or too long. For this group it seems that it might be appropriate to develop a shorter task, for which the results described above provide some alternatives (e.g., using exclusively syllable repetition, or a limited number of utterances such as the ones included in Table 3). Regarding the acoustic context, one potential problem was the presence of noise (e.g., from other speakers, electronic devices such as computers or air conditioners, cars) This might be particularly relevant for patients whose respiratory weakness can make their voice less audible than that of controls. However, the results indicate that this has not been an important limitation: the HN Scores were slightly lower in the patients with background noise than in the patients without background noise, which shows that the background of noise had a limited impact on the results. Altogether our results indicate that it is feasible to use mobile devices to make an automatic assessment of HN.

Our results point to some issues that require further exploration. In the first place, the results of this study suggest that patients and controls differed notably in how they produced syllables sequences and the sustained consonant /f/. However, we could not clarify whether these differences were due to the presence of nasal resonance or, alternatively, to adaptations in the articulatory patterns used to produce these sounds. Clarifying this issue might be most valuable to have better understanding of the speech characteristics of hypernasal speakers. In the second place, regarding the possibility of using mobile devices in speech evaluation, two limitations must be noted. In the first place, we did not carry a comparison between data obtained with our app and data obtained with other recording techniques. Unfortunately, due to the COVID-19 crisis, we were not able to obtain such data. In the second place, it should be emphasized that, in this study, we used devices of relatively high quality (i.e., iPhone and iPad). Thus, it is necessary to explore to what extent the results are the same independently of the recording tools used and whether or not the accuracy of the different classifiers remains equally high when using devices of a lower quality. Finally, future studies should explore whether or not the methodology used in this study can serve to grade the severity of HN and also to detect changes associated to medical and speech therapy treatments.

## 5. Conclusions

There are three main conclusions of the present study. The first one is that it is possible to use well known acoustic analysis and automatic classification algorithms to develop a HN detection tool based on running speech. The second conclusion is that the protocols for automatic evaluation of HN, like those used by human experts, should include a variety of utterance types. Finally, the third conclusion is that it is feasible today to use universally available tools such as mobile phones to evaluate HN.

**Author Contributions:** Conceptualization, I.M.-T.; methodology, I.M.-T. and R.B.-d.-A.; software, A.L.; validation, I.M.-T., A.L. and E.N.; formal analysis, I.M.-T. and E.N.; investigation, I.M.-T.; data curation, I.M.-T. and R.B.-d.-A.; writing—original draft preparation, I.M.-T. and A.L.; writing—review

and editing, R.B.-d.-A.; funding acquisition, I.M.-T. and E.N. All authors have read and agreed to the published version of the manuscript.

**Funding:** This research was funded by the Spanish MINISTERIO DE CIENCIA, INNOVACIÓN Y UNIVERSIDADES, grant number RTI2018-094846-B-I00 and JUNTA DE ANDALUCÍA (SPAIN), grant number UMA18-FEDERJA-021.

**Institutional Review Board Statement:** The study was conducted according to the guidelines of the Declaration of Helsinki and approved by the Ethics Committee of UNIVERSIDAD DE MÁLAGA (protocol code 14-2021-H, 12 April 2021).

**Informed Consent Statement:** Informed consent was obtained from all subjects involved in the study.

**Data Availability Statement:** The data presented in this study are openly available in GitHub repository https://github.com/Caliope-SpeechProcessingLab/Hypernasality (accessed on 1 May 2021). Raw audio data recorded is not available because it contains audio from underage patients.

**Acknowledgments:** The authors would like to thank all the participants and families for taking part in this study. We would like to acknowledge Mirta Palomares (Fundacion Gantz, Santiago de Chile), Franginett Quintana (Cuenca, Ecuador) and Wanda Meschian Coretti (Málaga, Spain) for their support to collect the data. Finally, we thank Wanda Meschian Coretti for her valuable help to annotate the full database.

**Conflicts of Interest:** The authors declare no conflict of interest.

## Appendix A. Instructions to Use the app ASICA

The app ASICA can be downloaded from Apple Store with no cost. It can be used in IOS type devices (iPhone and iPad) with at least iOS 12.4 or higher. A video tutorial (in Spanish) can be obtained from https://fb.watch/5JYIT32j6c/ (accessed on 1 May 2021).

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
