# Peer review of "Which Utterance Types Are Most Suitable to Detect Hypernasality Automatically?"

_applsci, doi:10.3390/app11198809_

Round 1

Reviewer 1 Report

REPORT

  • Please, refer to the methodology in the abstract.
  • Please, in the introduction, refer clearly to your research questions, hypothesis and objectives. You just say this at the end of the section but it is not enough:
  • This study explores to what extent utterances other than sustained vowels can be 64 used to detect hypernasality automatically. Our main aim is to identify which utterances, 65 if any, might be the most optimal ones for this task. In long term aim we aim to develop a 66 hypernasality detection tool that is accessible to a large audience, and particularly to cli-67 nicians without speech therapy expertise (e.g., pediatricians, otorhinolaryngologists, etc.) 68 In order to cope with this long-term objective, we decided to collect the data using a mo-69 bile app. Thus, a second aim of this study is to test to what point audio data obtained 70 using a mobile app can be used to evaluate hypernasality.
  • I think that section 1. Literature review should be section 2 and not a subsection inside the introduction. The same applies for section 1.2.
  • 2 Should be named as Methodology and have subsections: procedure, participants and context. I suggest that that section is joined to 2. Materials and Methods.
  • Table 1 on page 5 needs to be introduced in the previous paragraph.

DECISION: Accept with minor changes.

  • The conclusions need to be more elaborated.

Author Response

Response to Reviewer 1 Comments

Point 1: Please, refer to the methodology in the abstract.

Response 1: Please note that lines 18 to 23 in the abstract describe aspects of methodology. Due to space limitation it is not easy to add more information without missing other important details.

Point 2: Please, in the introduction, refer clearly to your research questions, hypothesis and objectives. You just say this at the end of the section, but it is not enough:

This study explores to what extent utterances other than sustained vowels can be 64 used to detect hypernasality automatically. Our main aim is to identify which utterances, 65 if any, might be the most optimal ones for this task. In long term aim we aim to develop a 66 hypernasality detection tool that is accessible to a large audience, and particularly to cli-67 nicians without speech therapy expertise (e.g., paediatricians, otorhinolaryngologists, etc.) 68 In order to cope with this long-term objective, we decided to collect the data using a mo-69 bile app. Thus, a second aim of this study is to test to what point audio data obtained 70 using a mobile app can be used to evaluate hypernasality.

Response 2: We have added anticipated results for both the research questions indicated above.  

Point 3: I think that section 1.1 Literature review should be section 2 and not a subsection inside the introduction. The same applies for section 1.2.

Response 3: Section 1.1 has been integrated in the Section 1. Introduction.

Section 1.2 has been moved to Section 2. Material and Methods.

Point 4: 1.2 Should be named as Methodology and have subsections: procedure, participants, and context. I suggest that that section is joined to 2. Materials and Methods.

Response 4: As suggested Section 1.1 and 1.2 have been modified.

Point 5: Table 1 on page 5 needs to be introduced in the previous paragraph.

Response 5: Added introduction to Table 1 at the end of the previous paragraph.

Point 6: The conclusions need to be more elaborated.

Response 6: Please note that rather than a discussion and conclusion section, we wrote a discussion section that elaborates the main ideas, and a summary conclusion that merely presents the key ideas. 

Reviewer 2 Report

The manuscript entitled “Which utterance types are most suitable to detect hypernasality automatically?” compared the hypernasality in different types of utterances and trained different classification algorithms. The results were interesting and provided an alternative viewpoint from previous studies. However, some of the issues (see below) need to be clarified.

Major:

  • The second research question is to examine whether an app like this would be suitable for nasality detection. However, there is no independent evidence to support this view. In other words, this app/technique should be compared with other techniques. If the effect is substantial, it should be reported as such.
  • (line 356) Please explain the composition of the speakers. Specifically, who are the 36 H and 49H? How are they different? Why two groups of healthy participants were included?
  • (line 363) Why Table 3 only lists the features from the 36P+36H database? Is there any reason or justification to choose this database from the others? 
  • (line 368) In subsection 3.3 Combining classifiers, the authors grouped different classifiers and compared the accuracy (as shown in Figure 5). However, it was not clear to me how these combinations were determined? Without explanations or rationales, these combination may be argued rather random, which may in turn have some significant impact on the final result interpretation. Please explain.
  • (lines 377-380) The authors stated that the best results are obtained in the two cases as in Figures 5e and 5f. I understand that Figure 5e would serve as a good combination since the histograms show high accuracy separating healthy speakers from patients. What I didn’t understand is why Figure 5f is a better case than Figure 5c? Is there any quantification method to determine which combination is better than the other? Please clarify.
  • (line 461 onwards) The authors conducted a separate measure for each participant by combining the results of multiple tests. The results to this new measure was not clearly reported. The text only reports that the results echoed Orozco-Arroyave et al’s findings. It’d be great to tell the readers what kinds of analyses have been performed and what the (statistical) results were.
  • (the paragraph of line 472) Here the authors mentioned two potential issues for the present study. As to the second issue, even with some background noise, the resultant accuracy was fairly high. The authors attributed this high accuracy to the successful hypernasality assessment using the mobile app. However, though it might be somehow unlikely, could it be the case that those background noise somehow correlate with the nasality signals and get picked up and analyzed. Can the authors talk about or even reject this possibility?

Minor

  • (line 63) Incomplete sentence.
  • (line 322) Please explain the bottom panel (i.e., the green lines) of Figure 2.
  • (lines 386 - 387). Please rephrase this sentence. I am not sure I am following the current form.
  • (lines 451-453) Incomplete sentence.
  • (line 460) it selected always —> it always selected
  • (line 474) related with —> related to

Author Response

Response to Reviewer 2 Comments

Please note that some of the comments of reviewer 2 refer to a previous version of our manuscript. That version was withdrawn because we found that there were important errors in the accuracy calculations that had an impact on the results and discussion. A new version was submitted on the 6th of July. Note also that this last version addresses some of the limitations mentioned by this reviewer. We apologize for the confusions created by withdrawing the previous version and submitting a new one.

Point 1: The second research question is to examine whether an app like this would be suitable for nasality detection. However, there is no independent evidence to support this view. In other words, this app/technique should be compared with other techniques. If the effect is substantial, it should be reported as such.

Response 1: We agree with the reviewer that this is a limitation of the present study. Unfortunately, due to the COVID-19 crisis, we were not able to obtain data using traditional recording techniques. We add a note to explain this limitation of our study (lines 578-9)

Point 2: (line 356) Please explain the composition of the speakers. Specifically, who are the 36 H and 49H? How are they different? Why were two groups of healthy participants included?

Response 2: Please note that in the present version we use only one balanced database composed of 39 healthy and 39 patient speakers. In the Method section we explain the motivation to select this set of participants.

Point 3: (line 363) Why Table 3 only lists the features from the 36P+36H database? Is there any reason or justification to choose this database from the others?  

Response 3: As noted above, in the present version we use only one database.

Point 4: (line 368) In subsection 3.3 Combining classifiers, the authors grouped different classifiers and compared the accuracy (as shown in Figure 5). However, it was not clear to me how these combinations were determined? Without explanations or rationales, this combination may be argued rather random, which may in turn have some significant impact on the result interpretation. Please explain.

Response 4: By combining multiple classifiers, we replicate the approach traditionally used by speech pathologists, in which the patient with multiple items, and the final score is the sum of the scores obtained with the individual items. Note also that the procedure to obtain these data is clarified in the present version (see HN Score description in section 2.6)

Point 5: (lines 377-380) The authors stated that the best results are obtained in the two cases as in Figures 5e and 5f. I understand that Figure 5e would serve as a good combination since the histograms show high accuracy separating healthy speakers from patients. What didn’t I understand is why Figure 5f is a better case than Figure 5c? Is there any quantification method to determine which combination is better than the other? Please clarify.

Response 5: Please note that in the present version Figures 5a-f have been replaced by Figures 5a-c., which were computed by selecting either all the utterances or a subset for which scores were above a minimum value. Lines 429-427 provide a quantitative and qualitative comparison of the results for controls and patients in the three figures 5a-c.

Point 6: (line 461 onwards) The authors conducted a separate measure for each participant by combining the results of multiple tests. The results to this new measure were not clearly reported. The text only reports that the results echoed Orozco-Arroyave et al’s findings. It’d be great to tell the readers what kinds of analyses have been performed and what the (statistical) results were.

Response 6: We believe that this issue has been addressed in the present manuscript. Please see section 2.6 (Method) and section 3.4. (Results)

Point 7: (the paragraph of line 472) Here the authors mentioned two potential issues for the present study. As to the second issue, even with some background noise, the resultant accuracy was fairly high. The authors attributed this high accuracy to the successful hypernasality assessment using the mobile app. However, though it might be somehow unlikely, could it be the case that those background noise somehow correlate with the nasality signals and get picked up and analysed. Can the authors talk about or even reject this possibility?

Response 7:

We thank the reviewer for this observation, we agree that this is a potential limitation of this study. In order to clarify the impact of background noise we have examined separately the results (for HN Score) in the patients with and without noise. This new paragraph has been added to the results (just before the discussion) to clarify this issue:

Given that the controls’ and the patients’ recordings differed in the amount of background noise, we analyzed whether this had any effect on the HN Scores. For this end, we divided the patients into two subgroups: those with background noise (N = 14) and those without background noise. Note that if the presence of background noise biased the results the patients with background noise might be classified as hypernasal more frequently that the remaining patients. However, the mean HN Score (7 + Sel) was slightly lower for the patients recorded with background noise than for those without noise (77%% vs. 83%). Thus, it seems that the presence of noise did not bias the results.

Point 8: (line 322) Please explain the bottom panel (i.e., the green lines) of Figure 2.

Response 8: Added explanation in Figure 2 text.

Point 9: (line 63) Incomplete sentence.

(lines 386 - 387). Please rephrase this sentence. I am not sure I am following the current form.

(lines 451-453) Incomplete sentence.

(line 460) it selected always —> it always selected

(line 474) related with —> related to

Response 9: All grammatical errors were corrected.

Round 2

Reviewer 2 Report

The manuscript has been revised considering all the points raised in the last round of review. However, there are some remaining issues to be taken care of. They are as follows.

  • (Section 3.3) What are T3, T5, …etc.? Are they S3, S5, as labeled in Table 1? Please be consistent.
  • (line 379 - 381) The authors said: “In the case of the RF and ANN tests, the results showed an increase in accuracy from one to two utterances, and a reduction with three or more utterances.” I didn’t follow where the reduction comes from. Please explain/elaborate. Similarly, please explain why the peak accuracy was obtained with three or four utterances in the case of SVM (lines 381-382).
  • (line 395) What is HN score (44[RMBDA1])? Why was it highlighted? Please clarify.
  • Figure 1: Hipernasal —> Hypernasal

Author Response

Point 1: (Section 3.3) What are T3, T5, …etc.? Are they S3, S5, as labeled in Table 1? Please be consistent.

Response 1: We replaced S3, S5, … with T3, T5, … to maintain consistency.

Point 2: The authors said: “In the case of the RF and ANN tests, the results showed an increase in accuracy from one to two utterances, and a reduction with three or more utterances.” I didn’t follow where the reduction comes from. Please explain/elaborate. Similarly, please explain why the peak accuracy was obtained with three or four utterances in the case of SVM (lines 381-382).

Response 2: We add an explanation in the discussion (Lines 518-522).

Note that this reduction in accuracy when the number of utterances increases is most possibly the result of the feature selection process, which did not take into account the correlation between the selected features. This may have led to select redundant features and to discard other features that might contribute to the classification problem.

Point 3: What is HN score (44[RMBDA1])? Why was it highlighted? Please clarify.

Response 3: HN score (44[RMBDA1]) is modified to HN Score (44). It was highlighted due to an error with the Track Changes tool.

Point 4: Figure 1: Hipernasal —> Hypernasal

Response 4: Changed in Figure 1.